# Interpreting Emergent Military Tactics in a General AlphaZero Framework

## Abstract

This paper presents an approach that combines AlphaZero with convolutional and transformer-based neural network architectures to learn strategies in battlefield-inspired gridworld games. These games are designed to balance realism with rapid outcomes, featuring multiple agents organized into competing teams. To encourage effective coordination among agents, we investigate different reward shaping methods and evaluate their impact on emergent teamwork. The learned strategies are analyzed on a tactical level, in an attempt to reveal insights into multi-agent collaboration and competitive behavior. In particular, the framework provides a testbed for studying how military-style strategies can emerge from self-play. Through a series of comparative studies, we further break down the contributions of architectural components and training methodologies to demonstrate the effectiveness of this approach for decision-making in dynamic adversarial settings.

## 1 Introduction

Modern military operations are defined by complexity, uncertainty, and the need for rapid decision-making in adversarial environments. Commanders must coordinate multiple units under conditions of resource scarcity, contested terrain, and evolving threats. As warfare increasingly involves decentralized teams operating in dynamic battlefields, the demand for tools that can simulate, analyze, and optimize tactical and strategic choices grows. Computational models make it possible to study military decision-making in controlled settings, allowing large-scale analysis of coordination, resource use, and planning against opponents.

Artificial intelligence methods, particularly reinforcement learning, have demonstrated their ability to generate adaptive strategies in competitive domains such as board games. Extending these capabilities into military-inspired settings enables the study of emergent tactics in scenarios that echo real-world battlefield constraints - limited ammunition, restricted weapon range, shrinking safe zones, and the necessity of holding strategic ground. These conditions mirror the challenges faced by military units tasked with balancing offensive aggression, defensive resilience, and the conservation of scarce resources while under constant threat from an intelligent adversary.

This work applies the AlphaZero (Silver et al., 2017; Schrittwieser et al., 2020) framework, augmented with modern neural network architectures, to battlefield-inspired gridworld games. These environments serve as testbeds for analyzing how military-style tactics, such as coordinated maneuvers, focused fire, and positional control, can emerge from self-play. Our study looks at how reward shaping and model selection affect strategy and how they impact performance. These findings help explain how AI agents can learn flexible, resilient decision-making.

## 2 Related Work

Several recent efforts have explored reinforcement learning in environments that combine spatial reasoning, adversarial dynamics, and team coordination. Gridworld combat simulations

have proven particularly effective as lightweight testbeds for emergent tactics. Peng et al. (2017) introduced multi-agent battle games where agents must coordinate movement and attack decisions under resource constraints. The StarCraft Multi-Agent Challenge (SMAC Vinyals et al. (2019)) further advanced this line of research by framing micromanagement tasks in real-time strategy battles, highlighting the difficulty of multi-agent coordination in partially observable, adversarial settings. Unlike these environments, our battlefield game is simpler and fully observable. This makes it easier to link learned behaviors to design and training choices, while still keeping key tactical features like limited ammo, shrinking safe zones, and contested objectives.

Reward shaping has been widely investigated as a means of improving coordination in adversarial games. Potential-based shaping Ng et al. (1999) has been shown to accelerate learning without altering the set of optimal policies, and has been applied to cooperative navigation and combat scenarios to encourage behaviors such as maintaining formation, focusing fire, or controlling strategic areas Nanxun et al. (2024). In gridworld-based combat, shaping terms related to distance-to-goal or team-health balance have been especially effective in promoting teamwork. Our study builds on this work by designing shaping functions that explicitly encourage cohesion, focus fire, and positional control, and by analyzing their impact on emergent strategies in a self-play AlphaZero framework.

Finally, evaluation of agent strength in competitive environments often relies on head-to-head competitions and Elo ratings Elo (1978), while methods like Sequential Probability Ratio Testing Wald (1945) provide statistical reliability.

## 3 METHODOLOGY

### 3.1 BATTLEFIELD GAME

To simulate a battlefield, we devised a simple, fully observable, zero-sum, gridworld-based game between two players. The game is played on a square grid of size $H \times W$ (with default value of $7 \times 7$) with two agents per team. At the start of play, Player 1's agents are placed randomly on the top row, and Player 2's agents on the bottom row, with a small number of obstacles randomly distributed to ensure partial cover while maintaining connectivity across the map. Each agent is initialized with a fixed amount of ammunition, and this resource is tracked explicitly in the state tensor alongside positional information, health, and center-control streak counters.

Gameplay unfolds in discrete turns, with each team selecting joint actions for its agents from a space of movement, waiting, or firing primitives. Because each individual agent can in general take 9 actions (4 move actions, 4 shoot actions, and 1 wait action), the dimension of the joint action space for 2 agents is $9^2 = 81$. Firing consumes one unit of ammo regardless of whether the shot hits, and agents that exhaust their supply cannot shoot further. The range of a shot is limited.

Line-of-sight rules decide if a shot hits, with obstacles and safe-zone edges blocking its trajectory. Furthermore, obstacles also block movement. The safe zone gradually shrinks inward at fixed intervals, instantly eliminating any agents that fall outside its bounds. This will induce the agents to move towards the goal state, and at the same time places a natural upper bound on the number of moves a game can have. A match ends when one or both teams lose all agents (due to firing or board shrinkage) or when the center has been held for the required number of steps. The outcome reward is zero-sum, with the victorious team receiving +1, the defeated team -1, and drawn games yielding zero. This occurs when all agents are eliminated.

These dynamics make the environment strategically rich. Limited ammunition forces players to weigh immediate attacks against saving resources for later, especially as the shrinking safe zone increases close encounters. Agents must coordinate offensive pressure, defensive positioning, and control of the center, all while managing a dwindling supply of shots. The combination of spatial control, resource management, and adversarial planning makes this game a challenging benchmark for studying multi-agent reinforcement learning under scarcity.

### 3.1.1 State Representation

The game state is represented as a 3-D tensor of size $H \times W \times 11$: planes 1 and 2 represent the agent locations of the current player, planes 3 and 4 those of the other player, plane 4 the position of the obstacles, planes 5 and 6 the remaining health points for resp. the current and other player, plane 7 the remaining steps, planes 8 and 9 the center-capture streak counters and planes 10 and 11 the remaining ammunition for current and other player. The advantage of representing everything from the perspective of the current player is that we can use the same neural network for both players; we only have to swap the corresponding planes. This improves training efficiency.

The environment's dynamics depend only on the current state $s_t$ and the players' action $a_t$. The next state $s_{t+1}$ is determined from these alone, meaning the Markov property holds:

$$P(s_{t+1} \mid s_t, a_t, s_{t-1}, a_{t-1}, \dots) \;=\; P(s_{t+1} \mid s_t, a_t).$$

While the game is fully observable and the dynamics are Markovian, we still augment the game state by passing extra information to the neural network. First, we can stack a configurable number of past states on top of the current canonical encoding, effectively giving the network access to short-term history in the form of additional planes. Second, we append broadcast planes (planes that uniformly filled with the same value across the whole board) that inject additional context: one plane encodes the normalized number of steps remaining in the episode, another marks whether the current position has already occurred in the same episode so serves as a repetition indicator, and a third encodes the normalized number of steps since the last damage event. These additional planes provide the network with information that is not strictly required for optimality, but can help it learn more efficiently by highlighting information that is otherwise implicit in the trajectory.

## 3.2 Training Loop

The training process follows the AlphaZero paradigm, consisting of a data generation phase and a training phase. An agent plays games against itself, generating training data at each turn. This data is then in the second phase used to update the weights of the neural network $f_\theta$.

The selection of the action to take during self-play is guided by a Monte Carlo Tree Search (MCTS (Kocsis and Szepesvári, 2006; Browne et al., 2012; Coulom, 2006; Gelly and Silver, 2011)), which is itself guided by a neural network $f_\theta$. The network takes a state representation $s$ and outputs a policy prior and a value estimate, $(p, v) = f_\theta(s)$. The MCTS algorithm uses these outputs to explore the search space. After a set number of simulations, the action to take is chosen based on the most visited child node of the game tree root $s_t$. This action is then executed and the game moves on to then new state $s_{t+1}$. This process continues until the episode is terminated.

After a fixed number of episodes, the training phase begins. The normalized visit counts form the training target $\pi$ for the policy head. The game's final outcome $z$ serves as the training target for the value head. These $(s, \pi, z)$ tuples are stored in a replay buffer, from which batches are sampled to train the network $f_\theta$.

### 3.2.1 MCTS

Monte-Carlo Tree Search is a tree-traversal algorithm that operates in three distinct phases. Firstly, during the *selection* phase, the next node in the tree-traversal process is selected with the help of the PUCT (polynomial upper confidence trees) selection rule. At each node, this rule selects the action that maximizes a balance between exploitation - represented by the current action-value estimates $Q(s, a)$ - and exploration, implemented by an upper-confidence bound term that depends on the network's policy prior and the node's visit counts. Concretely, the action $a$ selected at state $s$ is

$$a \;=\; \arg\max_{a'} \left[ Q(s, a') \;+\; c_{\text{puct}}\, p(a'|s) \, \frac{\sqrt{\sum_b N(s, b)}}{1 + N(s, a')} \right],$$

where $p(a|s)$ is the prior probability of action $a$ under the neural network's policy, $N(s,a)$ is the visit count for that state-action pair, and $c_{\text{puct}}$ is an exploration constant. This rule ensures that moves with higher action-value estimates or stronger priors are favored, while still allocating visits to less explored actions.

Secondly, we enter the *expansion* phase: when the search encounters a leaf node that has not yet been visited, the neural network is called to provide (i) a policy distribution over all legal actions of this node, and (ii) a scalar value prediction. The policy is used to initialize the prior probabilities of the node's outgoing edges. The value estimate of this new leaf node is then backed up along the path in the third phase called *backpropagation*: state-action visit counts are incremented, and action-value estimates $Q(s,a)$ are updated as running averages of the backed-up values.

Several enhancements make this process more effective in AlphaZero. The search tree is reused between moves: after a real action is selected and played on the board, the corresponding child becomes the new root, preserving past simulations and avoiding recomputation. To promote diversity in self-play, we add Dirichlet noise to the root priors so the search explores different plausible moves. Moreover, during self-play, moves are sampled proportionally to visit counts rather than chosen deterministically, which broadens the training distribution.

### 3.2.2 REWARD SHAPING

A Markov Decision Process (MDP - Puterman (2014)) is the standard mathematical framework to model decision-making under uncertainty over time and formalizes how a player interacts with an environment. Formally, an MDP is a 5-tuple:

$$\mathcal{M} = (\mathcal{S}, \mathcal{A}, P, R, \gamma)$$

with a $\mathcal{S}$ the set of states, $\mathcal{A}$ the set of possible actions and $P(s' \mid s, a)$ the probability of landing in state $s'$ after taking action $a$ in state $s$. The reward $R(s, a, s')$ is the immediate feedback an agent receives after moving from $s$ to $s'$ via $a$. In our game, this reward is only rewarded at the end of the game. It is +1 for winning, -1 for losing and 0 for a draw. Furthermore, an MDP has a discount factor $\gamma \in [0, 1]$ which determines how much future rewards matter compared to immediate ones.

The goal of an MDP is to find a policy $\pi(a_t|s_t)$ such that the discounted accumulated reward

$$G_t = \sum_{k=0}^{\infty} \gamma^k R(s_{t+k}, a_{t+k}, s_{t+k+1})$$

is maximized. This policy is the optimal policy $\pi^*$.

Reward shaping (Ng et al., 1999) augments the raw MDP reward with an additional signal derived from a potential function $\phi(s)$ over states. Instead of altering the true terminal reward $r_t$, it adds a discounted difference of potentials between successive states $s_t$ and $s_{t+1}$:

$$r_t^{\text{shaped}} = r_t + \gamma \phi(s_{t+1}) - \phi(s_t)$$

One can show (see Ng et al. (1999)) that adding this potential difference $d_t = \gamma \phi(s_{t+1}) - \phi(s_t)$ to the original sparse reward does not change the optimal policy of the MDP.

The strength of AlphaZero comes from combining large-scale search with deep function approximation, without the need for hand-crafted shaping terms. However, potential-based reward shaping can be deliberately added to speed up learning when rewards are delayed or sparse. We apply reward-shaping in two locations in the training loop:

1. When a MCTS simulation steps from state $s_t$ to next state $s_{t+1}$ for the current actor, the code computes a local potential difference $d_t = \gamma \phi(s_{t+1}) - \phi(s_t)$ and stores it along the path[1]. During backpropagation, these differences are accumulated to form an additive term that is added to the terminal outcome $z$ before updating each edge's total value. This directly impacts MCTS search.

2. Separately, when a self-play game ends and training examples are finalized, the value training targets can be shaped offline. Trajectories $(s_t, a_t, s_{t+1})$ are recorded,

---

[1]It is important to remember that in state $s_{t+1}$ the player who moves is the next player, hence the potential must be adjusted accordingly.

with the acting player at each step. The algorithm recomputes the same $d_t = \gamma\,\phi(s_{t+1}) - \phi(s_t)$, discounts, and accumulates them backward. This cumulative potential term then added to the usual terminal result $z$ to yield the per-time-step training label $z_t = z + \sum_{k \geq t} \gamma^{k-t}\,d_k$. This directly impacts model training.

The following shaped rewards can be applied:

1. The *team-health advantage* $\phi_1$ measures the difference in the number of alive agents between both players.

2. A *center control reward* $\phi_2$ measures the best (minimum) inverse distance among each team's agents to the board center.

3. A *goal-proximity* bonus averages inverse distance to the center over all alive agents and rewards broad, coordinated progress toward the capture square.

4. To reflect the shrinking safe zone, a *border-safety term* $\phi_3$ rewards being farther from the shrinking boundary.

5. Tactical pressure is captured by a *line-of-sight threat* $\phi_4$ feature that checks whether any of our agents has an unobstructed view to any opponent within weapon range.

6. Two coordination bonuses drive within-team synergy: *cohesion* ($\phi_5$) favors teammates being closer together, and *focus-fire* ($\phi_6$) rewards both agents choosing the same nearest opponent as their most attractive target.

All terms are combined linearly with weights $w_i$ and clipped to $[-1, 1]$:

$$\phi(s_t) = \mathrm{clip}\Big(\sum_{i=1}^{6} w_i \phi_i(s_t), -1, 1\Big)$$

### 3.2.3 Model acceptance

Model acceptance plays a central role in AlphaZero-style training, as it determines whether the network being trained replaces the current baseline used for generating self-play data. Without a principled acceptance criterion, training can drift, either by promoting weak models that degrade performance or by stagnating because promising models are never adopted.

In our implementation, two acceptance strategies are available. The first is a simple threshold test, where a candidate model plays a fixed number of evaluation games against the previous snapshot, and is accepted if its win rate exceeds a preset threshold. While straightforward, this approach can be noisy, especially with limited evaluation games. To address this, the code also supports pool-based acceptance using Sequential Probability Ratio Testing (SPRT - (Wald, 1945)). Here, the candidate is tested against a pool of the last accepted models, with outcomes evaluated using a likelihood ratio test between a null hypothesis of equal strength ($p_0 = 0.50$) and an alternative of improvement ($p_1 = 0.55$). This provides a statistically rigorous mechanism for early acceptance or rejection. If SPRT remains inconclusive after a cap on evaluation games, the decision falls back to the Wilson lower confidence bound (LCB) (Wilson, 1927) of the observed win rate, accepting only if the bound clears a safety threshold.

### 3.3 Neural Networks

The neural network in AlphaZero is trained directly from the outcomes of the MCTS process. For each position encountered during self-play, the search produces an improved policy target $\pi$ in the form of the normalized visit counts over actions at the root, and a value target $z$ that reflects the eventual game outcome (optionally augmented with potential-based shaping). These targets are paired with the canonical board representation to form training examples $(s_t, \pi_t, z_t)$. The network is then updated by minimizing a composite loss: a cross-entropy term aligning its policy head with the visit-count distribution, a mean-squared error term aligning its value head with the target outcome, and a regularization penalty to prevent overfitting:

$$\mathcal{L} = (z - v)^2 - \pi^T \log(p) + c\|\theta\|^2$$

We use both convolutional neural network (He et al., 2016) and Transformer-based architectures Vaswani et al. (2017). Details can be found in appendix A. Both types have a common core, a value head to predict value $v$ and a policy head to predict vector $p$. The transformer network can also process action tokens, which are extra sequence elements representing legal moves. These additional tokens are added to the flattened board state.

### 3.4 ELO RATING

To systematically compare different models, we rely on Elo rating (Elo, 1978) calculations. Elo is a logistic, match-by-match rating system that adjusts players' ratings based on performance relative to expectation. For players $A$ and $B$ with ratings $R_A$ and $R_B$, the expected score for $A$ is

$$E_A = 1/(1 + 10^{(R_B - R_A)/400})$$

After a game the ratings are updated according to

$$R_A \leftarrow R_A + K(S_A - E_A) \text{ and } R_B \leftarrow R_B + K(S_B - E_B)$$

where $S_{A,B} \in \{1, \frac{1}{2}, 0\}$ for respectively a win, draw or loss and $K$ is a coefficient that controls volatility. We set $K = 32$. A 400 point rating difference advantage to 10-to-1 odds of winning.

## 4 EXPERIMENTS

### 4.1 STRATEGIES

We cannot strictly speak of "learned strategies" as the training process does not directly optimize for tactical sequences of moves. Instead, what is learned is (i) an estimate of the expected outcome of a given board state through the value head, and (ii) a prior probability distribution over actions that guides MCTS via the policy head.

To illustrate typical gameplay, figure 1 shows a duel between two players (red and blue) on a $7 \times 7$ grid. The game play is the result of using the AlphaZero algorithm with a transformer network, trained over 1000 iterations with reward shaping. One iteration is a cycle of data generation with MCTS and learning of the neural network. Initially, the agents are positioned on opposite sides of the board, while the central objective square is marked by a yellow cross. In steps 1 and 2, both players advance toward the central region (note that intermediate game steps are omitted for clarity). In subfigure 2, the board begins to shrink (grey squares), but all agents have moved within the safe zone. By subfigure 4, the blue team has successfully positioned its agents to defend or capture the center, whereas the red team has already lost one agent due to the second shrinkage. In the final stage (subfigure 5), the blue player occupies the central square. Because shooting is restricted to horizontal and vertical lines of sight, the remaining red agent is unable to contest this position, resulting in victory for the blue team.

Very often though, teams lose one of their agents early on the game - mostly due to board shrinkage - such that the end game is typically one-on-one. This endgame then reduces to a waiting game: both remaining agents keep the goal square under fire, prohibiting the other player from occupying the goal square. Unless mistakes are made, these games almost always end in a draw. The most important conclusion is that players don't learn to coordinate the behavior of their agents. To accomplish coordination, more refined reward shaping is probably needed.

### 4.2 IMPACT OF REWARD SHAPING

We trained two transformer networks with 4 layers and 4 heads per layer. The first network was trained with reward shaping (both during MCTS and model training - see section 3.2.2) while the other one was trained without. Figure 2 shows how the loss (policy, value and total) evolves during training - the different graphs have been smoothed with an exponential moving average (EMA) for clarity. As can be seen, the use of shaping has little to no impact

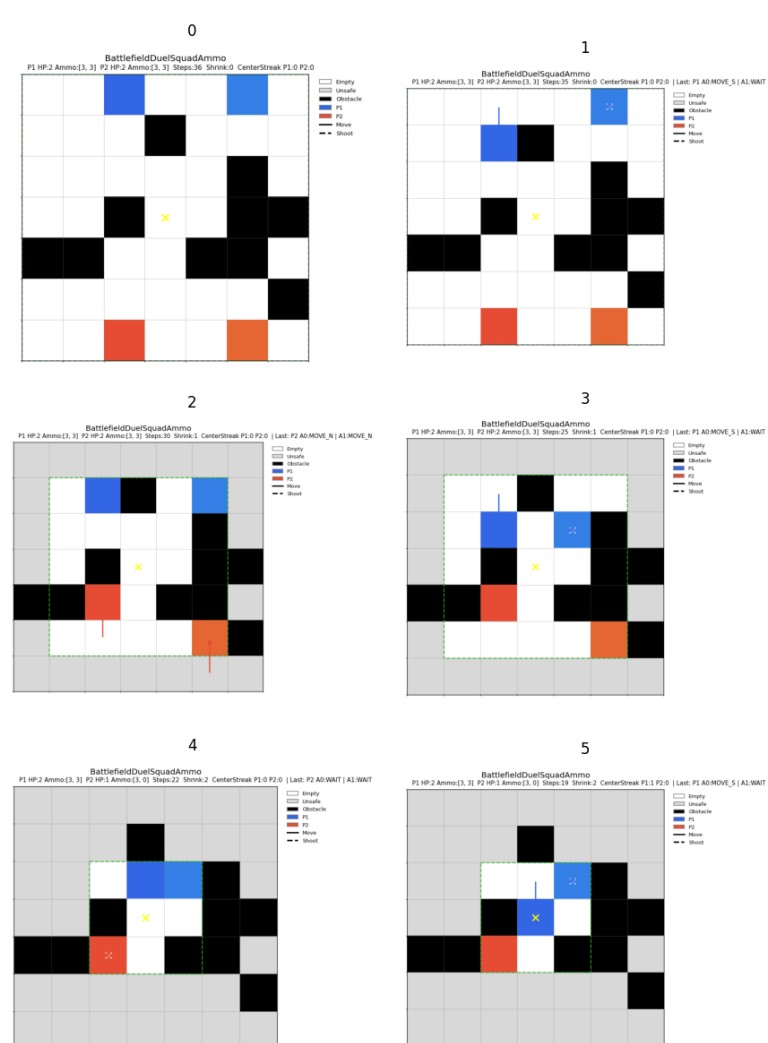

Figure 1: Gameplay example

on the training loss.

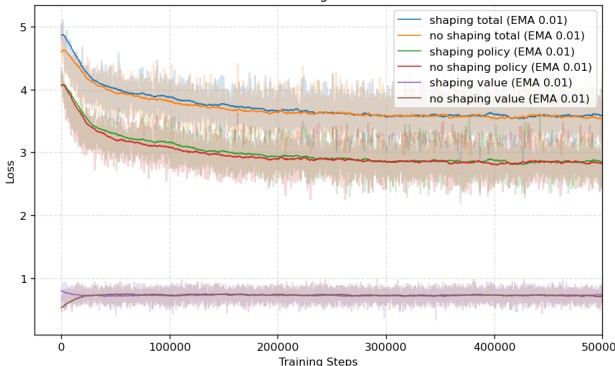

Figure 2: Loss graph with and without shaping

| Rank | Model | Elo |
|------|-------|-----|
| 1 | shaping100 | 1649.5 |
| 2 | shaping400 | 1582.5 |
| 3 | noshaping400 | 1571.3 |
| 4 | shaping500 | 1554.5 |
| 5 | noshaping100 | 1493.9 |
| 6 | noshaping500 | 1474.2 |
| 7 | random | 1174.0 |

Table 1: Elo ranking - reward shaping

We also compared the resulting models based on their performance using the Elo metric, for models after 100, 400 and 500 training iterations for both training with and without

reward shaping - the games themselves for Elo calculations were played without any reward shaping (this should have been learned during training). The results are shown in table 1. The models learned with reward shaping perform better in general. However, training duration doesn't seem to improve performance (something we also notice in section 4.3).

### 4.3 IMPACT OF ACCEPTANCE THRESHOLD

As mentioned previously, model acceptance or rejection has a profound impact on the behavior of the algorithm: it determines whether the network being trained replaces the current baseline used for generating self-play data, which might lead to favoring weak models with low performance. Because the model essentially plays against itself, this weakness can remain hidden during training. This is why more involved acceptance mechanisms like SPRT are used. In this section however, we study two training scenario's where acceptance is purely based playing a fixed number of evaluation games against the previous model snapshot and having a win rate higher than some threshold. In the first scenario, this threshold is low (60%), in the second scenario, the threshold is put at 90%. In both scenario's, a transformer network with 4 layers and 4 heads per layer is trained.

Figures 3 and 4 shows the impact of the acceptance threshold. The total training loss in figure 3 is slightly lower when acceptance is easier, because the model adapts more rapidly to the provided data. The win rates of the new model (figure 4(a)) are similar but decreasing, which has a profound impact of the acceptance rate, but while the win rate stays steadily above 60%, it regularly drops below 90% such that the second scenario has a much lower (and decreasing) average acceptance rate (figure 4(b)).

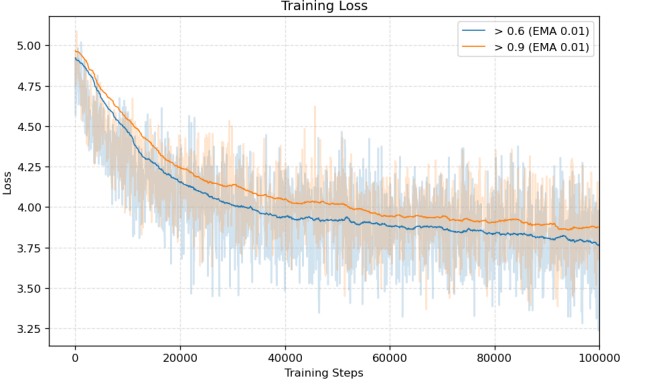

| Rank | Model | Elo |
|------|-------|-----|
| 1 | hard100 | 1660.0 |
| 2 | hard200 | 1613.1 |
| 3 | hard300 | 1578.6 |
| 4 | hard500 | 1560.3 |
| 5 | standard200 | 1551.9 |
| 6 | hard400 | 1515.7 |
| 7 | standard400 | 1512.3 |
| 8 | standard300 | 1498.4 |
| 9 | standard500 | 1468.7 |
| 10 | standard100 | 1464.0 |
| 11 | random | 1076.9 |

Figure 3: Impact of different acceptance ratio - Training Loss

Table 2: Elo ranking - acceptance ratio

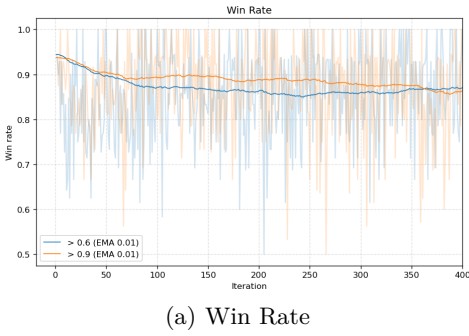

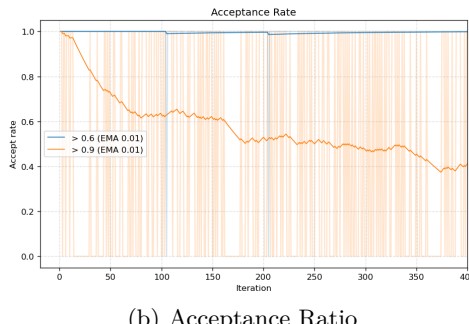

(a) Win Rate

(b) Acceptance Ratio

Figure 4: Impact of different acceptance ratio. (a) Win rate (b) Acceptance rate

These loss curves don't say anything about the performance of the learned models. To compare them, we selected 5 models from the standard 60% acceptance scenario (after respectively 100, 200, 300, 400 and 500 iterations (`standard100` to `standard500`) and 5

models from the 90% acceptance scenario (`hard100` to `hard500`). As a baseline, we also put in an agent that selects action at random. We computed the different Elo scores (with 8 interactions between all models) - see table 2. From this table, we can draw two main conclusions: (i) a higher acceptance threshold leads to better performing models (the top 3 is occupied by models learned with the 90% acceptance threshold) and (ii) - and more worrying - models that were learned later in the learning process don't necessarily perform better then earlier models.

### 4.4 IMPACT OF ARCHITECTURE

We compare 3 different types of neural networks as described in section 3.3, who each trained for 1000 iterations: a convolutional neural net (`cnn`), a transformer network with action tokens (`trans01`) and a standard transformer network without action tokens (`trans02`). The Elo rating of these networks and a random agent were computed (table 3) where each agent ran 1000 MCTS simulations before choosing an action.

| Rank | Model | Elo |
|------|---------|--------|
| 1 | trans01 | 1662.4 |
| 2 | cnn | 1606.2 |
| 3 | trans02 | 1527.6 |
| 4 | random | 1203.8 |

Table 3: Elo ranking - architectures

This table shows that including action tokens in the transformer architecture significantly improves performance; this architecture outperforms both the CNN and the vanilla transformer.

## 5 CONCLUSION

This study shows how an AlphaZero framework, using both convolutional and transformer models, can develop tactical and strategic behavior in battlefield-like settings. By adding features such as limited ammunition, shrinking safe zones, and contested objectives, the environment recreated important aspects of combat and attempted to force agents to balance attack, defense, and teamwork. The experiments demonstrated that, through self-play, agents can learn competitive strategies without explicit programming.

The investigation of reward shaping, model acceptance thresholds, and architectural variations revealed not only their technical impact on training performance but also their practical implications for military-style coordination.

The results show that AI-based testbeds can help explore how tactics and strategies emerge. Even in simplified gridworld settings, the framework captures key dynamics like resource competition, spatial control, and adaptation. Future work could broaden the range of agents, environments, and goals to study how AI might inform or challenge human approaches to command and control. Another useful addition would be the use of league training Vinyals et al. (2019) as an alternative to SPRT to create models that consistently become better as training progresses.

## 6 REPRODUCIBILITY STATEMENT & LLM USAGE

In order to ensure reproducibility, the necessary python code is provided as a zip file together with this submission. The authors acknowledge the use of LLM's for help with writing this paper.

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

# A  Neural Networks Structure

## A.1  Convolutional neural network

As a baseline, we use a convolutional neural network (He et al., 2016) tailored to the AlphaZero framework, with a structure that mirrors the classical "torso + dual heads" design. The input is the canonical board tensor of shape $(C, H, W)$ with the number of channels $C$ determined by the chosen state representation (see section 3.1.1) . A first convolutional layer with 64 filters of size $3 \times 3$ (stride 1, padding 1) followed by batch normalization and ReLU activation projects the raw input into a higher-dimensional feature space. This is followed by three residual blocks, each consisting of two $3 \times 3$ convolutions with batch normalization and a skip connection, producing features of the same channel dimension.

From this shared body, the network branches into two specialized heads. The policy head begins with a $1 \times 1$ convolution that reduces the feature channels to a small number (2 by default), followed by batch normalization, ReLU, and flattening, then a fully connected layer that outputs logits over all legal actions. The value head also starts with a $1 \times 1$ convolution reducing the channels to 1, followed by batch normalization, ReLU, flattening, and two fully connected layers: the first maps to the hidden dimension (equal to the torso channel size), and the second reduces to a single scalar. A hyperbolic tangent activation bounds this value in $[-1, 1]$.

## A.2  Transformer network

Alongside the convolutional network, we also use a Transformer (Vaswani et al., 2017)-based architecture where the convolutions are replaced with attention layers. The input is still the canonical board state tensor, but instead of being processed spatially by convolutions, it is flattened into a sequence of tokens: each board cell corresponds to a token, with its input features given by the stacked state planes at that location. To this sequence, the model prepends a dedicated [CLS] token, whose embedding is trained to summarize the global state. After the sequence passes through a stack of Transformer encoder layers (consisting of multi-head self-attention and feedforward blocks with residuals and normalization), the [CLS] representation serves as a pooled embedding that captures global context across the entire board.

To preserve spatial structure after flattening, the model adds a positional encoding to each token. By default this is a learnable embedding of row and column indices, ensuring that the network can distinguish otherwise identical features that occur in different parts of the grid. The model can also process action tokens, which are extra sequence elements representing legal moves. These are constructed with simple features such as normalized row and column coordinates of the target action, along with mask indicators of legality. When included, the action tokens attend jointly with board tokens, giving the network a direct channel to reason about action-specific features in context with the current position.

The network outputs again bifurcate into a policy head and a value head. The policy head typically reads from the action tokens (or from per-cell embeddings mapped back to the grid) and produces logits over all legal actions. The value head uses the [CLS] embedding, passed through one or more dense layers and a final tanh activation, to predict the expected game outcome in $[-1, 1]$. Thus, while structurally different, the Transformer and CNN architectures play analogous roles: both learn shared state representations and feed them into dual heads, but the Transformer introduces more flexible, attention-based modeling of spatial and action dependencies, with the [CLS] token acting as a learned global summary.

