# OpenReview forum: "Interpreting Emergent Military Tactics in a General AlphaZero Framework"
_ICLR.cc/2026/Conference — ICLR 2026 Conference Withdrawn Submission_

### Official Review · Reviewer_Wo4W · 2025-10-30

**Soundness:** 3
**Presentation:** 3
**Contribution:** 2
**Rating:** 2
**Confidence:** 3

**Summary:**

The authors describe a new multi-agent reinforcement learning environment designed to mimic military tactics. The environment simulates real-world battlefield constraints like ammunition, weapon range etc. The environment is clearly described in the paper along with the game play and state representation. Experiments on the environment were performed investigating different policy architecture, reward shaping etc.

**Strengths:**

The environment is very clearly defined and there are potentials of establishing as a new environment.
Effects of different training factors like reward shaping are thorough

**Weaknesses:**

The paper lacks novelty in terms of algorithm or modeling improvements.
Techniques described in the paper are very well established in the literature.

The environment could potentially be beneficial in the military space, there are limit novelty as a benchmark.
There were limited comparisons with existing multi-agent test benches like Google Research Football.
Furthermore, even the SMAC seems to be more complex and diverse compared to the test bench described.

**Questions:**

* Instead of randomly assigning players' agent on rows, could there be specific scenarios? This can test the generalizations of the policies
* Have you compared with simple baselines like DQN?

---

### Official Review · Reviewer_ZwTf · 2025-11-01

**Soundness:** 2
**Presentation:** 2
**Contribution:** 1
**Rating:** 0
**Confidence:** 5

**Summary:**

The authors introduce a new battlefield gridworld environment and test AlphaZero-inspired automated game playing approaches in it. The authors present the environment and then test the AlphaZero-inspired approaches. They find that they largely work. The authors present a great deal of outputs to characterize the performance in this environment.

**Strengths:**

The primary strength is the new environment. This is fairly novel though there are a number of similar environments, such as MicroRTS and Generals.io. However, there is no originality to the AI approaches presented in the paper. The quality of the work is standard, this is the typical approach any researchers would take to applying these approaches to a new environment. The clarity of the work is very high in terms of both the environment and the AlphaZero implementation. The significance is unfortunately low, as there are similar existing environments and there aren't any surprising results for AlphaZero-like approaches in such environments.

**Weaknesses:**

The fundamental problem with this work is that the contributions are not relevant to an AI venue. There is a great deal of work on automated game playing for similar environments [1,2,3]. None of the approaches applied in this environment are novel in the literature.

To make this work relevant to an AI venue I'd recommend that the authors consider an analysis that compares their environment to similar existing environments to clarify how they are similar and different, and present experiments to support these claims.

1. Bhatia, Aaditya, et al. "Generally Genius: A Generals. io Agent Development and Data Collection Framework." Proceedings of the AAAI Conference on Artificial Intelligence and Interactive Digital Entertainment. Vol. 19. No. 1. 2023.
2. Straka, Matej, and Martin Schmid. "Artificial Generals Intelligence: Mastering Generals. io with Reinforcement Learning." arXiv preprint arXiv:2507.06825 (2025).
3. Ontañón, Santiago, et al. "The first microrts artificial intelligence competition." AI Magazine 39.1 (2018): 75-83.

**Questions:**

1. How do the authors relate their environment to similar environments like MicroRTS and Generals.io?

---

### Official Review · Reviewer_T7VF · 2025-11-01

**Soundness:** 1
**Presentation:** 2
**Contribution:** 1
**Rating:** 2
**Confidence:** 3

**Summary:**

This work focuses on applying self-play and MCTS-based RL methods, previously proposed and successfully applied in Alpha-zero (Silver et. al., 2017), in a grid-world setting. The authors particularly investigated the impact of reward-shaping, model acceptance scheme, and network architecture on the performance of the learned agent.

**Strengths:**

- The Battlefield Game environment could serve as an additional benchmark for multi-agent or MCTS-based research. Overall, the provided source code runs and includes proper docstrings and type hints, albeit with incomplete coverage.

**Weaknesses:**

In my opinion, the paper proposes no novelty. This work instead serves as a toy(-ish) example of MCTS and self-play approaches.
- The lack of standard benchmarks in the empirical studies (even if the goal is to train an agent solely in the proposed environment) makes the analysis of network architectures (e.g., Transformers vs. CNNs) and reward shaping methods hard to interpret and impossible to compare with prior work; as a result, the conclusions are largely anecdotal and lack external validity.

**Questions:**

- Were the agents on each team run in a decentralized manner?
- (Minor) Figure 1 is difficult to read.

**Details Of Ethics Concerns:**

I am concerned about the target application of the paper. On multiple occasions, the authors discuss the implications of the paper’s observations for a military setting. I am not sure whether this is considered an ethical issue, but I felt the need to flag it.

---

### Official Review · Reviewer_fXEi · 2025-11-01

**Soundness:** 2
**Presentation:** 2
**Contribution:** 1
**Rating:** 2
**Confidence:** 4

**Summary:**

This paper proposes an adaptation of the AlphaZero framework to a battlefield gridworld environment, aiming to study the emergence of military-style tactics via self-play reinforcement learning.
The authors design a simplified environment featuring limited ammunition, shrinking safe zones, and center-control objectives.
Based on this, this work investigates the effects of different reward shaping terms and model architectures, and analyze the emergent tactical behavior.

**Strengths:**

1. The paper clearly explains the AlphaZero training pipeline, state representation, and reward shaping mechanism.
2. As a small-scale testbed, it could be useful for education or for preliminary exploration of the generation of gaming behavior.

**Weaknesses:**

1. The entire methodology is a direct and unmodified (or small modidified) application of AlphaZero to a small toy gridworld. Common techniques such as reward shaping and acceptance threshold tuning are employed, but the paper introduces no novel algorithmic mechanism, theoretical insight, or analytical framework.
2. There are no baseline comparisons with established MARL frameworks. Moreover, no statistical analysis, ablation, or environmental diversity is presented, and all experiments are restricted to a single 7×7 gridworld.
3. Overall, the paper reads more like an engineering report documenting the implementation of AlphaZero with minor adjustments, rather than a research contribution offering new understanding or advancement.

**Questions:**

see weaknesses

---

### Note · Authors · 2026-01-26

I have read and agree with the venue's withdrawal policy on behalf of myself and my co-authors.

---

### Meta-Review · Area_Chair_rjpw · 2026-01-06

**Summary:**

All the reviewers agree that the contribution and novelty of this paper are limited. It is not an AI paper but an reproduction of AI approach in other domains.

**Reviewer Concerns:**

Not addressed.

**Reviewer Scores:**

0 0 0 0

---

### Decision · Program_Chairs · 2026-01-26

Reject